# Disentangling the causes of temporal variation in the opportunity for sexual selection

Rômulo Carleial ®[1,2] ✉, Tommaso Pizzari[1], David S. Richardson[3] & Grant C. McDonald ®[4] ✉

In principle, temporal fluctuations in the potential for sexual selection can be estimated as changes in intrasexual variance in reproductive success (i.e. the opportunity for selection). However, we know little about how opportunity measures vary over time, and the extent to which such dynamics are affected by stochasticity. We use published mating data from multiple species to investigate temporal variation in the opportunity for sexual selection. First, we show that the opportunity for precopulatory sexual selection typically declines over successive days in both sexes and shorter sampling periods lead to substantial overestimates. Second, by utilising randomised null models, we also find that these dynamics are largely explained by an accumulation of random matings, but that intrasexual competition may slow temporal declines. Third, using data from a red junglefowl (*Gallus gallus*) population, we show that declines in precopulatory measures over a breeding period were mirrored by declines in the opportunity for both postcopulatory and total sexual selection. Collectively, we show that variance-based metrics of selection change rapidly, are highly sensitive to sampling durations, and likely lead to substantial misinterpretation if used as indicators of sexual selection. However, simulations can begin to disentangle stochastic variation from biological mechanisms.

Driven by intrasexual competition over reproductive opportunities, sexual selection is a powerful evolutionary mechanism, which varies widely over space and time[1–9]. Understanding the causes and consequences of such variation is an enduring challenge in evolutionary biology. Temporal fluctuations in sexual selection can affect processes that ultimately determine net selection across a reproductive period, with repercussions for rates of adaptation, the evolution of alternative reproductive tactics, the maintenance of variation in competitive traits, plasticity in sexual phenotypes and dynamism in evolutionary sex roles[10–13]. Temporal fluctuations may arise as a consequence of variation in the ecological[4,10,14–20] or social environment[8,11,21–26], drastically changing mating dynamics[8,27]. While some of these environmental changes may be long-term[28,29], others can be more rapid, e.g.

within a breeding season[9,11]. Sexual receptivity, courtship effort, mating propensity and mate choice preferences often change sharply in matters of few hours or days[30–35]. While studies have assessed changes in sexual selection across multiple breeding seasons or years[4,10,36–40], less is known about the potential for rapid fluctuations in sexual selection over these much shorter, but still biologically relevant, scales.

The strength of sexual selection is measured by regressing measures of reproductive success over given phenotypic measures among members of the same sex within a population (i.e. selection gradients[41]). However, identifying and measuring the traits (e.g. ornament expression, courtship behaviour or sperm quality) causally related to the outcome of intrasexual competition and variation in reproductive success is often very challenging[42–46]. Thus, a widely used

[1]Department of Zoology, Edward Grey Institute, University of Oxford, Oxford OX1 3SZ, UK. [2]Science Directorate, Royal Botanic Gardens, Kew, Richmond TW9 3AE, UK. [3]School of Biological Sciences, University of East Anglia, Norwich, UK. [4]Department of Ecology, University of Veterinary Medicine Budapest, Budapest 1077, Hungary. ✉e-mail: r.carleial@kew.org; grant.mcdonald@univet.hu

alternative approach estimates the potential for – rather than the strength of – sexual selection in a population as standardised intra-sexual variance in reproductive success. This metric, known as the 'opportunity for selection', captures the maximum potential strength of sexual selection in a population even when little is known about the phenotypic traits that mediate intrasexual competition[47,48]. In addition, the opportunity for selection can be utilised to dissect the potential for sexual selection into components driven by variance in the number of sexual partners (precopulatory sexual selection) and in the proportion of gametes fertilised (postcopulatory sexual selection)[49]. The value of these variance-based metrics however, remains debated, largely due to their inability to distinguish between variation that arises due to intrasexual competition and variation in reproductive success due to stochastic processes[42,43,50,51]. For example, Klug et al.[42] showed that the number of unmated individuals tends to increase as the operational sex ratio becomes more biased, so the opportunity for sexual selection will also increase even if individuals mate at random. Moreover, variance-based metrics may be particularly sensitive to sampling effort (i.e. the period of time over which sampling occurs), and because they measure patterns of variance relative to mean reproductive success, these metrics may be strongly impacted by temporal increases in mean reproductive success[50]. For example, sampling for an insufficient period may overestimate opportunity measures by inflating the number of non-mating individuals[52–54]. Alternatively, temporal changes in the opportunity for sexual selection may be driven by competitive biological processes. In principle, behavioural shifts towards increased monogamy, promiscuity, mate monopolisation or increased sperm competition over time may all dynamically change the opportunity for sexual selection. However, because the opportunity for sexual selection only measures the max-imum potential for sexual selection and does not discriminate between competitive and stochastic processes[42,43,51], there is a risk that sys-tematic changes in this metric over time may be mistakenly inter-preted as changes in actual sexual selection. An important step toward understanding temporal dynamics of sexual selection is therefore to resolve how variance-based metrics change over a reproductive period and the extent to which these patterns are reflective of biological *versus* methodological and stochastic processes (e.g. variation in the sampling period; random mating). However, currently there is little information on how the opportunity for sexual selection behaves over time, and the methodological and biological processes that contribute to such dynamics. Understanding how variance-based metrics behave through time in relation to deterministic and stochastic processes is important because variance in reproductive success is a prerequisite for sexual selection.

Here, we address this knowledge gap by utilising fine-scale tem-poral data, at the resolution of a day, on sexual behaviour across dif-ferent vertebrate and invertebrate species. First, we characterise temporal changes in the opportunity for precopulatory sexual selection over cumulative days in males and females. Second, we assess the extent to which temporal changes in the opportunity for precopulatory sexual selection arise as a consequence of stochastic processes under the null expectation of random mating (as opposed to deterministic processes such as behavioural shifts in mating dynamics)[42]. Third, we establish the impact of sampling effort by comparing observed opportunity esti-mates based on cumulative patterns of mating (i.e. where mating suc-cess for each individual is summed across all preceding days and thus represents the maximum possible strength of selection over a sampling period up to and including a given day) with 'instantaneous' estimates calculated independently each day. Finally, using available parentage data from a population of red junglefowl (*Gallus gallus*) - a poly-gynandrous bird - we explore to what extent patterns identified for precopulatory sexual selection are also reflected in the opportunity for postcopulatory sexual selection (i.e. standardised variance in paternity share) and in the total opportunity for selection on reproductive success (i.e. standardised variance in reproductive success). We show that the opportunity for sexual selection varies sharply over time, is highly sensitive to sampling duration, and may lead to substantial mis-interpretation if used as an indicator of sexual selection. However, simulations should help to disentangle variation arising from stochastic and deterministic processes.

## Results

### Dynamics in the opportunity for precopulatory sexual selection across species

Our results demonstrate broadly consistent decreases over time (i.e. successive days within a breeding period) in the cumulative opportunity for precopulatory sexual selection on mating success ($I_M$) within both males and females across species (Fig. 1, Table 1) - with one exception. In the socially monogamous jackdaw (*Corvus monedula*) cumulative $I_M$ did not significantly change over time in males but increased towards the end of the study period in females (Fig. 1, Table 1).

We evaluated whether temporal trends in cumulative $I_M$ could be driven by the 1st day of an observation period. This is because in experimental studies, day 1 can correspond to the day when previously sex-segregated individuals are introduced to the opposite sex, which may influence mating behaviour (e.g.[55,56]). To test the effect of day 1, we used permutations to randomly shuffle the day order of mating events in our data (e.g. mating events on day 10 were assigned to day 1, and vice versa). Temporal trends in cumulative $I_M$ were qualitatively similar across permutations (Supplementary Fig. 1) suggesting patterns are not solely caused by the specific sequence of events occurring over days, but rather by the accumulation of mating events over time. This accumulation is exemplified by the increase in mean mating success across males and females of each species, although full saturation of the mating matrix was not observed in any species (i.e. in no group did all possible pairs of males and females copulate) (Supplemen-tary Fig. 2).

Null expectations for cumulative $I_M$ that assume random patterns of mating showed qualitatively similar tendencies to decline over time as compared to observed cumulative $I_M$ for both males and females (Fig. 1), demonstrating that a general temporal decline in $I_M$ is expected as a consequence of the random accumulation of mating events over time. However, if changes in variance across species are at least par-tially driven by competitive biological processes (e.g. mate mono-polisation), we should expect observed opportunity values to deviate from null expectations (i.e. the 95% range of the simulated values). Consistent with this, we found that observed cumulative $I_M$ values in some species were quantitatively higher than null expectations as observation periods progressed for both males and females, with significant deviations often starting from relatively early in the obser-vation period (Fig. 1). Two species showed sex-specific patterns. In squirrel monkeys (*Saimiri oerstedi*), while the observed cumulative $I_M$ for males was higher than null expectations, female observed cumu-lative $I_M$ consistently failed to deviate from null expectations. Similarly, cumulative $I_M$ for male strawberry poison-dart frogs (*Dendrobates pumilio*) was higher than null expectations towards the end of the observational period whereas female cumulative $I_M$ did not differ from null expectations.

Daily instantaneous estimates of $I_M$ did not show a temporal decline and consistently overestimated $I_M$ when compared to cumu-lative estimates, particularly later in an observational period (Fig.1, Table 1). These results indicate that the opportunity for sexual selec-tion can be substantially overestimated when measured over a restricted period.

### Dynamics in the opportunity for total and postcopulatory sexual selection in red junglefowl

The cumulative opportunity for total sexual selection ($I_T$) in both male and female red junglefowl consistently declined over time at a

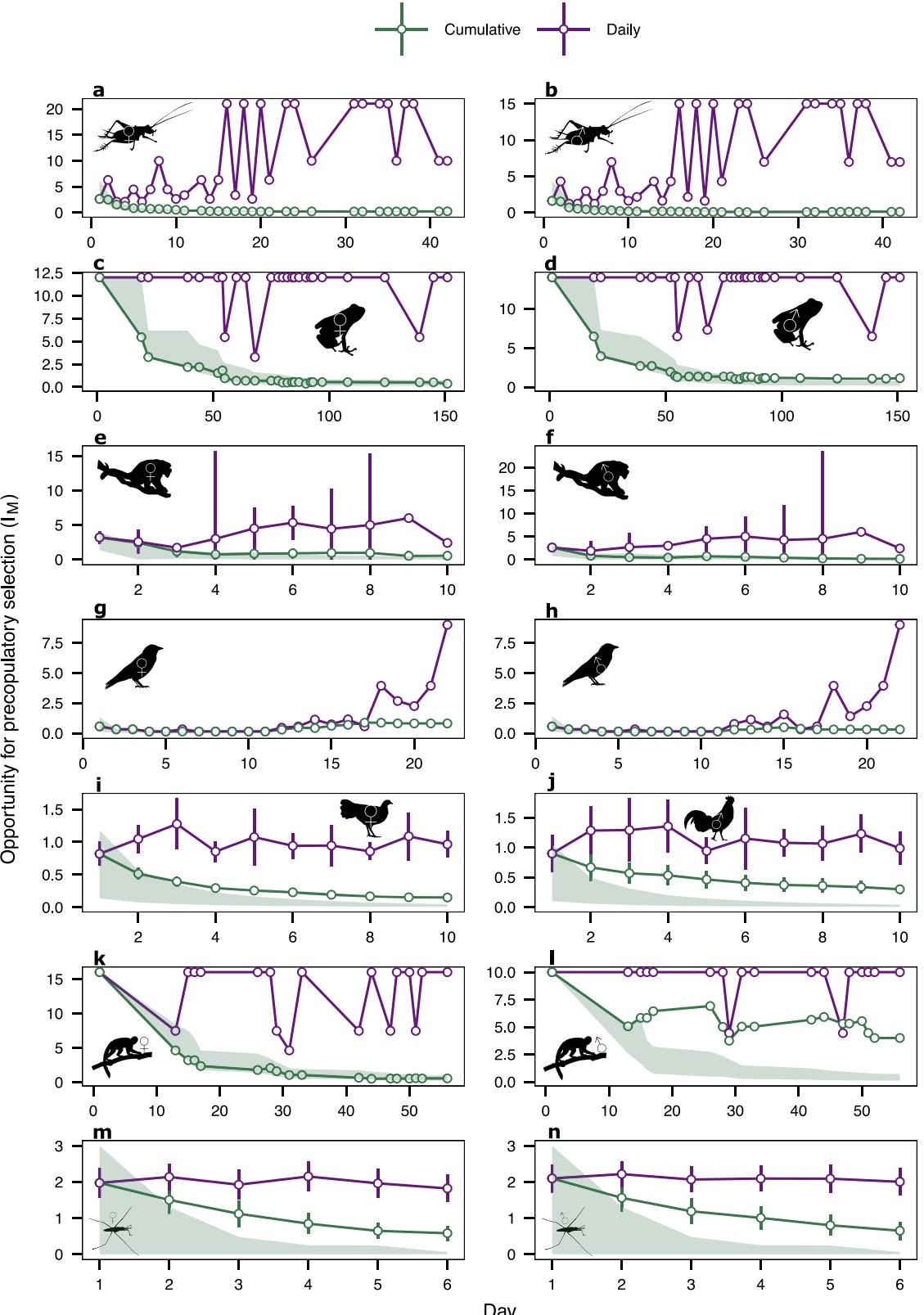

decelerating rate (Fig. 2a, b, Table 2). A similar pattern was observed when considering variance in the fecundity of males' sexual partners ($I_N$), which also decreased over time at a decelerating rate (Fig. 2c, Table 2). Finally, the opportunity for postcopulatory sexual selection ($I_P$) on male paternity share (P) also decreased over successive days, but this time linearly (Fig. 2d).

Similar to the opportunity for precopulatory sexual selection on mating success ($I_M$), simulations assuming random mating and fertilisations showed that measures of the total opportunity for sexual selection ($I_T$) and its constituent components ($I_N$ and $I_P$) decreased over time (Fig. 2). In comparison, observed estimates were often higher than null expectations, particularly later in a trial (Fig. 2). However, in

**Fig. 1 | Short-term temporal dynamics in the opportunity for precopulatory sexual selection across species.** Mean (95% c.i) values for the opportunity for precopulatory sexual selection ($I_M$) across vertebrate and invertebrates species over time (days) during a breeding period. Females (♀) and males (♂) are represented on the left and right panels, respectively. Species from top to bottom represent: **a**, **b** Hawaiian swordtail crickets[79] (*Laupala cerasina*) n groups = 1. **c**, **d** Strawberry poison-dart frogs[78] (*Dendrobates pumilio*) n groups = 1; **e**, **f** howler monkeys[77] (*Alouatta caraya*) n groups = 6. **g**, **h** Jackdaws[80] (*Corvus monedula*) n groups = 1. **i**, **j** Red junglefowl[56,66,67,81] (*Gallus gallus*) n groups = 20. **k**, **l** Squirrel monkeys[66] (*Saimiri oerstedi*) n groups = 1. **m**, **n** Water striders[8] (*Aquarius remigis*) n groups = 40. Green points show $I_M$ values calculated cumulatively and represent the maximum potential strength of precopulatory sexual selection on a given day taking into account patterns of mating over all preceding time units, and purple points show instantaneous values where opportunity for precopulatory sexual selection is assessed independently for each time unit (day). Bars represent the 95% confidence intervals (95% c.i) for studies with replicate groups. The lower range of c.i were capped at 0 to exclude negative values. Green shaded areas represent expectations for cumulative measures of $I_M$ based on 10,000 random mating simulations (i.e. the 95% range of simulated estimates). Observed cumulative values with c.i lying within shaded areas were considered not to differ from null expectations. Source data are provided as a Source Data file.

females, observed $I_T$ did not deviate from null expectations at any point in the trial (Fig. 2a). In males, observed $I_T$ was higher than null expectations from the 8th day of sampling onwards (Fig. 2b). Similarly, observed male $I_N$ and $I_P$ were higher than null expectations from the 8th day (Fig. 2c, d).

Daily instantaneous measures led to a significant overestimation of most opportunity estimates in both sexes. Daily estimates of $I_T$ were significantly higher than cumulative estimates (Fig. 2a, b, Table 2). Similarly, standardised variation in the fecundity of females mated by a male ($I_N$) was consistently higher when estimated using daily *versus* cumulative measures (Fig. 2c, Table 2). In contrast, we found no significant overall difference between daily and cumulative estimates of the potential for postcopulatory sexual selection ($I_P$) on males (Fig. 2d, Table 2).

## Discussion

Determining the causes and consequences of temporal variation in mating and reproductive success is an important step in understanding the evolutionary diversity of reproductive strategies. However, temporal variation within reproductive events is often overlooked in sexual selection studies, and the value of variance-based metrics being used as a proxy to assess temporal patterns of sexual selection has not been considered. Here, we utilise the standardised variance in mating success (i.e. the opportunity for precopulatory sexual selection, $I_M$) – commonly used as a proxy for the strength of sexual selection – to evaluate temporal changes in the potential for precopulatory sexual selection over a reproductive period across different animal species. We show that (i) the opportunity for sexual selection may decline rapidly over the course of just a few days of sampling for both females and males, (ii) observed temporal decreases are broadly similar to those expected under random mating, although some studies show patterns of non-random mating which slow the decrease in opportunity over time, and (iii) estimates are consistently overestimated if calculated over restricted snapshots of time (i.e. daily instantaneous measures). Finally, using detailed parentage data in a polygynandrous bird, we show that consistent declines in precopulatory measures were mirrored by temporal patterns in the opportunity for postcopulatory and total sexual selection. These results reinforce previous suggestions that the opportunity for sexual selection should not be used as a proxy of actual sexual selection, but that the use of null models can provide insight into how random versus deterministic processes are expected to influence mating dynamics.

We observed a reduction in the opportunity for precopulatory sexual selection ($I_M$) over time, as males and females accumulate more mating partners. Previous studies have suggested that increases in polyandry in already moderately polygynandrous populations may reduce $I_M$ in males by saturating the mating matrix and eroding intrasexual variation in mating success[55,57]. For example, in a rare study of short-term temporal variation in sexual selection over eight weeks in the hermaphroditic pond snail (*Lymnaea stagnalis*), variation in mating success in groups of five individuals was completely eroded in the first weeks of mating as all individuals had mated with all possible partners[7]. While mating matrices did not fully saturate in any of the

species in our present study, we demonstrate that $I_M$ can erode over time even in species with strong social hierarchies, where social dominance is presumed to confer male control over access to mating opportunities and maintain reproductive skew (e.g. red junglefowl). This erosion could, in principle, occur if subordinate males engaging in alternative mating strategies[58,59] are able to progressively attain mating partners over extended periods. This conclusion holds true for red junglefowl since it has been repeatedly shown that subordinate males achieve some mating success through sexual coercion[60]. Alternatively, females may increasingly express preferences to mate with novel males, or otherwise phenotypically different males over time, resulting in temporal reduction in mating skew[61,62]. Similar mechanisms could explain our observed temporal declines in $I_M$ in females, although this pattern has - to our knowledge - not been considered previously. In contrast, studies in socially monogamous populations indicate that some polyandry may increase variance in male mating success by allowing some males to attain more than one mate (i.e. via extra-pair copulations)[63]. In accordance with this, we showed that in the socially monogamous jackdaws, $I_M$ in females increased towards the end of the sampling period, which could be caused by female extra-pair mating or sequential polyandry at later stages of the reproductive season[64,65]. Future studies should explicitly compare the erosion of $I_M$ across an expanded range of socially monogamous species with systems characterised by strongly skewed mating patterns (e.g. lek and polygynous harem forming species) to understand the generality of such different temporal trajectories in opportunities for sexual selection. More importantly, such comparisons across different mating systems should include longitudinal measures of actual selection gradients[42] to explore how net patterns of phenotypic sexual selection are shaped by cumulative patterns of mating over time, such as via changes in female selectivity or alternative mating tactics.

Previous studies have criticised the use of opportunity estimates as reliable indicators of the strength of sexual selection, since variance in reproductive success may be partially driven by random mating or sampling issues alone (e.g.[42,43,50,51]). Our results strongly support this criticism, since temporal trends in opportunity estimates derived from simulations assuming random mating (all studies) and fertilisations (red junglefowl) were, to a large degree, qualitatively similar to observed values. Moreover, empirical estimates of opportunity often did not differ from simulated estimates, particularly early in a study period. One caveat is that our simulations had many simplifying assumptions and did not include detailed aspects of mating behaviour specific to each species. Moreover, in some field studies not all individuals were screened simultaneously, whereas in our simulations all individuals were assumed to be present on each sampling day. Despite these important limitations, we show that comparing null expectations with observed measures should help disentangling temporal trends driven by random mating from possible biological causes. For example, observed $I_M$ in male squirrel monkeys was higher than the simulated range as early as the third day of the sampled period, which can be explained by the largest male obtaining 70% of observed copulations in the original study[66]. This scenario, in which mate monopolisation is substantial, has been previously identified as one of the

**Table 1 | Temporal patterns in the opportunity for precopulatory sexual selection ($I_M$) across successive days (Day) calculated using two different methods (i.e. cumulative or daily instantaneous mating success) for multiple animal species**

| Predictor | Estimate±s.e | Statistic | p | Estimate±s.e | Statistic | p |
|---|---|---|---|---|---|---|
| **Strawberry poison dart frog (Dendrobates pumilio)**[A] | | | | | | |
| | | Males | | | Females | |
| Intercept [Cumulative] | 0.2 ± 0.05 | - | - | −0.46 ± 0.07 | - | - |
| Method [Daily] | 2.37 ± 0.05 | $F_{1,48} = 1144.0$ | **< 0.001** | 2.83 ± 0.09 | $F_{1,48} = 895.3$ | **< 0.001** |
| Day | −0.01 ± 0.00 | $F_{1,48} = 156.9$ | **< 0.001** | −0.02 ± 0.00 | $F_{1,48} = 191.5$ | **< 0.001** |
| Day² | 0 ± 0.00 | $F_{1,48} = 79.3$ | **< 0.001** | 0 ± 0.00 | $F_{1,48} = 61.9$ | **< 0.001** |
| Method [Daily] *Day | 0.01 ± 0.00 | $F_{1,48} = 71.0$ | **< 0.001** | 0.02 ± 0.00 | $F_{1,48} = 90.9$ | **< 0.001** |
| Method [Daily] *Day² | 0 ± 0.00 | $F_{1,48} = 41.6$ | **< 0.001** | 0 ± 0.00 | $F_{1,48} = 29.9$ | **< 0.001** |
| **Jackdaw (Corvus monedula)**[A] | | | | | | |
| | | Males | | | Females | |
| Intercept [Cumulative] | −1.59 ± 0.17 | - | - | −1.59 ± 0.14 | - | - |
| Method [Daily] | 0.33 ± 0.24 | $F_{1,38} = 1.8$ | 0.191 | 0.47 ± 0.15 | $F_{1,39} = 9.8$ | **0.003** |
| Day | 0.02 ± 0.02 | $F_{1,38} = 1.9$ | 0.180 | 0.08 ± 0.02 | $F_{1,39} = 24.0$ | **< 0.001** |
| Day² | 0.01 ± 0.00 | $F_{1,38} = 2.5$ | 0.125 | 0.01 ± 0.00 | $F_{1,39} = 10.0$ | **< 0.001** |
| Method [Daily] *Day | 0.12 ± 0.03 | $F_{1,38} = 23.1$ | **< 0.001** | 0.08 ± 0.02 | $F_{1,39} = 39.1$ | **0.003** |
| Method [Daily] *Day² | 0.01 ± 0.00 | $F_{1,38} = 5.9$ | **0.020** | - | - | - |
| **Hawaiian sword tail cricket (Laupala cerasina)**[A] | | | | | | |
| | | Males | | | Females | |
| Intercept [Cumulative] | −2.48 ± 0.13 | - | - | −1.53 ± 0.12 | - | - |
| Method [Daily] | 4.36 ± 0.19 | $F_{1,58} = 349.7$ | **< 0.001** | 3.80 ± 0.17 | $F_{1,58} = 501.1$ | **< 0.001** |
| Day | −0.07 ± 0.01 | $F_{1,58} = 82.7$ | **< 0.001** | −0.06 ± 0.01 | $F_{1,58} = 90.0$ | **< 0.001** |
| Day² | 0.00 ± 0.00 | $F_{1,58} = 39.0$ | **< 0.001** | 0.00 ± 0.00 | $F_{1,58} = 33.6$ | **< 0.001** |
| Method [Daily] *Day | 0.13 ± 0.01 | $F_{1,58} = 144.2$ | **< 0.001** | 0.12 ± 0.01 | $F_{1,58} = 153.4$ | **< 0.001** |
| Method [Daily] *Day² | −0.01 ± 0.00 | $F_{1,58} = 37.4$ | **< 0.001** | −0.00 ± 0.00 | $F_{1,58} = 33.9$ | **< 0.001** |
| **Squirrel monkey (Saimiri oerstedi)**[A] | | | | | | |
| | | Males | | | Females | |
| Intercept [Cumulative] | 1.85 ± 0.10 | $F_{1,33} = 365.2$ | **< 0.001** | 0.08 ± 0.10 | $F_{1,31} = 0.7$ | 0.410 |
| Method [Daily] | 0.53 ± 0.08 | $F_{1,33} = 46.7$ | **< 0.001** | 2.24 ± 0.11 | $F_{1,31} = 416.7$ | **< 0.001** |
| Day | −0.01 ± 0.00 | $F_{1,33} = 4.3$ | **0.046** | −0.05 ± 0.00 | $F_{1,31} = 116.0$ | **< 0.001** |
| Day² | - | - | - | 0.00 ± 0.00 | $F_{1,31} = 8.7$ | **0.006** |
| Method [Daily] *Day | - | - | - | 0.06 ± 0.01 | $F_{1,31} = 64.4$ | **< 0.001** |
| **Howler monkey (Alouatta caraya)**[B] | | | | | | |
| | | Males | | | Females | |
| Intercept [Cumulative] | 0.70 ± 0.08 | - | - | 0.91 ± 0.08 | - | - |
| Method [Daily] | 0.68 ± 0.10 | $F_{1,52} = 48.3$ | **< 0.001** | 0.60 ± 0.08 | $F_{1,58} = 51.0$ | **< 0.001** |
| Day | −0.08 ± 0.03 | $F_{1,55} = 0.1$ | 0.742 | −0.09 ± 0.02 | $F_{1,60} = 0.8$ | 0.376 |
| Method [Daily] *Day | 0.15 ± 0.04 | $F_{1,52} = 17.6$ | **< 0.001** | 0.16 ± 0.03 | $F_{1,58} = 25.9$ | **< 0.001** |
| **Water strider (Aquarius remegis)**[B] | | | | | | |
| | | Males | | | Females | |
| Intercept [Cumulative] | 0.67 ± 0.05 | - | - | 0.64 ± 0.05 | - | - |
| Method [Daily] | 0.40 ± 0.04 | $F_{1,357} = 112.0$ | **< 0.001** | 0.41 ± 0.04 | $F_{1,357} = 131.5$ | **< 0.001** |
| Day | −0.12 ± 0.02 | $F_{1,358} = 31.8$ | **< 0.001** | −0.12 ± 0.01 | $F_{1,358} = 40.0$ | **< 0.001** |
| Method [Daily] *Day | 0.11 ± 0.02 | $F_{1,355} = 24.0$ | **< 0.001** | 0.11 ± 0.02 | $F_{1,356} = 28.1$ | **< 0.001** |
| **Red junglefowl (Gallus gallus)**[B] | | | | | | |
| | | Males | | | Females | |
| Intercept [Cumulative] | −1.04 ± 0.12 | - | - | −1.75 ± 0.12 | - | - |
| Method [Daily] | 1.00 ± 0.06 | $F_{1,375} = 309.5$ | **< 0.001** | 1.87 ± 0.07 | $F_{1,374} = 813.9$ | **< 0.001** |
| Day | −0.11 ± 0.01 | $F_{1,375} = 64.2$ | **< 0.001** | −0.19 ± 0.01 | $F_{1,374} = 80.6$ | **< 0.001** |
| Day² | 0.01 ± 0.00 | $F_{1,375} = 1.5$ | 0.219 | 0.02 ± 0.00 | $F_{1,374} = 4.8$ | **0.029** |
| Method [Daily] *Day | 0.11 ± 0.01 | $F_{1,375} = 72.6$ | **< 0.001** | 0.24 ± 0.02 | $F_{1,374} = 248.6$ | **< 0.001** |
| Method [Daily] *Day² | −0.01 ± 0.01 | $F_{1,375} = 7.1$ | **0.008** | −0.03 ± 0.01 | $F_{1,374} = 23.7$ | **< 0.001** |

The results for males and females are shown on the left and right, respectively. Models were run separately per each species and sex. We used linear regression (LR) analyses for studies with a single observation per day (A), and linear mixed models (LMMs) including a random effect of group identity for studies with more than one replicate (B). Response variables ($I_M$) were log-transformed to meet model assumptions, and continuous predictors (i.e. Day, Day²) were centred to facilitate interpretation of fixed effects coefficients. Statistically significant results ($p < 0.05$) are in bold. F-statistics, $p$-values and degrees of freedom for LMMs were calculated using the Satterthwaite approximation.

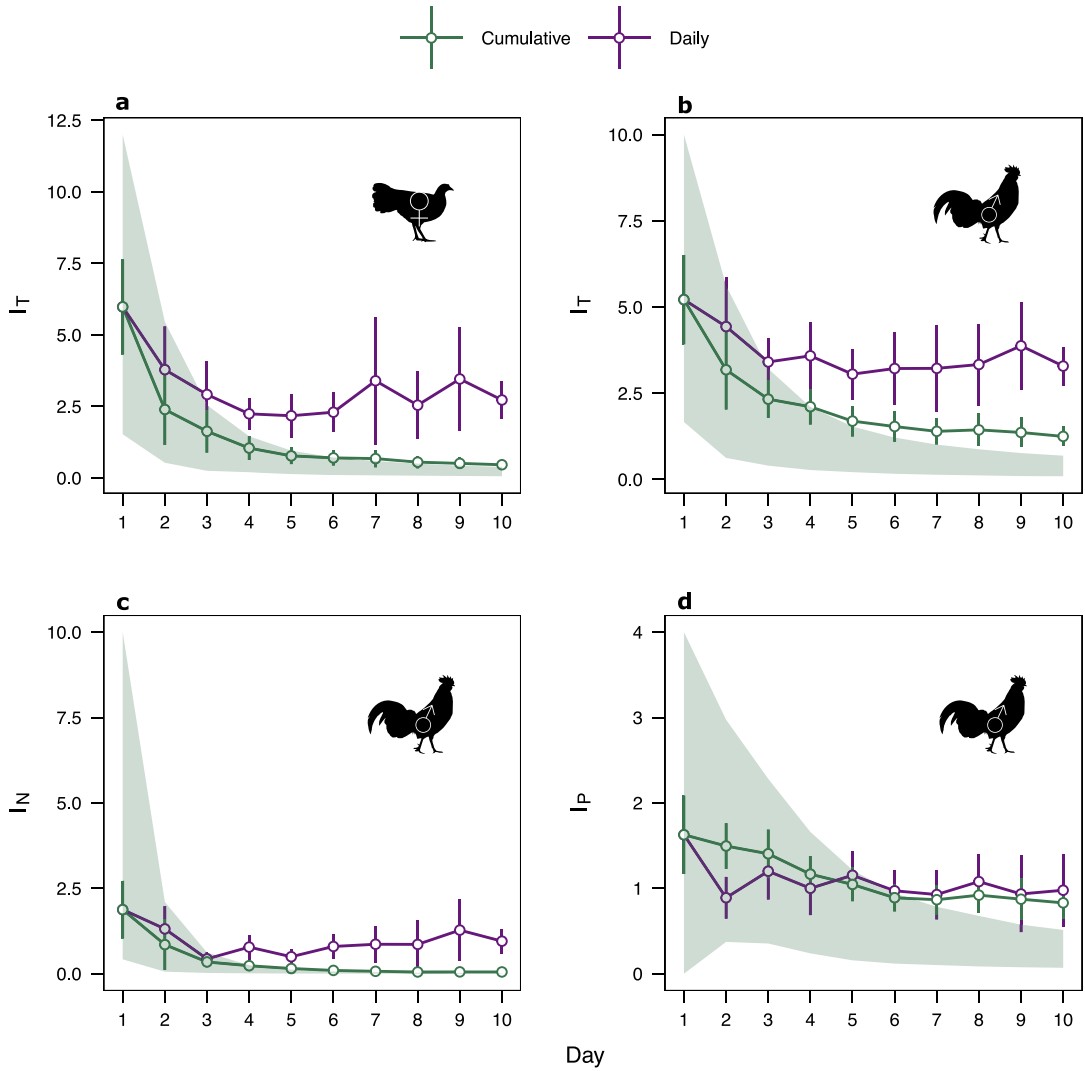

**Fig. 2 | Short-term temporal dynamics in multiple opportunity for selection episodes in a polygynandrous bird.** Patterns of opportunity estimates across different components of reproductive success in male and female red junglefowl (*Gallus gallus*) *n* groups = 20. Points represent observed mean opportunity estimates, bars represent 95% confidence intervals (95% c.i.) and point colour reflects the sampling method (green: cumulative; purple: daily). Green shaded areas represent expectations for cumulative measures of opportunity for selection based on 10,000 random mating simulations (i.e. the 95% range of simulated estimates).

Observed cumulative values with c.i. lying within shaded areas were considered not to differ from null expectations. Females (♀): **a** Standardised variance in reproductive success (opportunity for total selection, $I_T$). Males (♂): **b** Opportunity for total selection ($I_T$), **c** Standardised variance in partner's fecundity (opportunity for sexual selection on partner fecundity, $I_N$), **d** Standardised variance in paternity share (opportunity for postcopulatory sexual selection, $I_P$). Source data are provided as a Source Data file.

few cases where opportunity estimates may correlate with actual sexual selection[42]. Similarly, the higher $I_M$ than expected by chance in male red junglefowl is likely driven by a combination of male-male competition and female choice. Previous research with this red junglefowl dataset has shown that younger and more aggressive individuals tend to secure more mating partners and copulate more frequently[56,67]. Moreover, observed values of the opportunity for postcopulatory sexual selection ($I_P$) and total selection ($I_T$) deviated from null expectations much later in the trial than $I_M$. This suggests that consistent skew in mating success may take time to translate into biases in paternity share and reproductive success. This is because sperm from more successful males still must compete with rival sperm from earlier inseminations, which must be used up, or leaked, from the female reproductive tract before any consistent advantage is observed[68,69]. Together, these results reinforce previous suggestions indicating that null models should be included in sexual selection studies to adequately disentangle variance arising from competitive

versus stochastic causes[42,43]. For simulations to be biologically relevant, assumptions should be carefully defined in order to reflect particularities of the study system e.g. mating frequencies and sperm storage duration.

We also identified other concerns associated with using $I_M$ as a surrogate for the strength of sexual selection. Our comparisons between cumulative and daily instantaneous measures of opportunity for selection indicate that even small differences in sampling periods of a few days can lead to very different conclusions regarding the potential for sexual selection. Daily instantaneous measures consistently overestimated variance in components of reproductive success, suggesting that studies using relatively short periods of sampling relative to the overall length of the breeding event must be careful when interpreting results, particularly those pointing to a strong opportunity for sexual selection. Future studies aimed at quantifying the opportunity for sexual selection should devise the sampling duration carefully and be aware of the strong potential to overestimate

**Table 2 | Temporal patterns in the opportunity for total selection ($I_T$) and its components ($I_N$, $I_P$), across successive days (Day) calculated using two different methods (i.e. cumulative or daily instantaneous reproductive success) across multiple replicate groups of red junglefowl (*Gallus gallus*)**

| Response | Predictors | Estimate±s.e | Statistic | *p* |
|---|---|---|---|---|
| Females | | | | |
| $I_T$ | Intercept [Cumulative] | −0.5 ± 0.12 | - | - |
| | Method [Daily] | 1.06 ± 0.05 | $F_{1,369} = 477.2$ | **< 0.001** |
| | Day | −0.24 ± 0.01 | $F_{1,369} = 313.2$ | **< 0.001** |
| | Day$^2$ | 0.04 ± 0.00 | $F_{1,369} = 109.5$ | **< 0.001** |
| | Method [Daily] *Day | 0.18 ± 0.02 | $F_{1,369} = 112.2$ | **< 0.001** |
| Males | | | | |
| $I_T$ | Intercept [Cumulative] | 0.35 ± 0.09 | - | - |
| | Method [Daily] | 0.61 ± 0.04 | $F_{1,369} = 210.5$ | **< 0.001** |
| | Day | −0.14 ± 0.01 | $F_{1,369} = 141.6$ | **< 0.001** |
| | Day$^2$ | 0.02 ± 0.00 | $F_{1,369} = 54.5$ | **< 0.001** |
| | Method [Daily] *Day | 0.1 ± 0.01 | $F_{1,369} = 45.0$ | **< 0.001** |
| $I_N$ | Intercept [Cumulative] | 0.09 ± 0.05 | - | - |
| | Method [Daily] | 0.33 ± 0.03 | $F_{1,369} = 110.1$ | **< 0.001** |
| | Day | −0.07 ± 0.01 | $F_{1,369} = 62.1$ | **< 0.001** |
| | Day$^2$ | 0.02 ± 0.00 | $F_{1,369} = 59.5$ | **< 0.001** |
| | Method [Daily] *Day | 0.06 ± 0.01 | $F_{1,369} = 27.9$ | **< 0.001** |
| $I_P$ | Intercept [Cumulative] | 0.71 ± 0.03 | - | - |
| | Method [Daily] | −0.04 ± 0.03 | $F_{1,359} = 2.1$ | 0.144 |
| | Day | −0.03 ± 0.01 | $F_{1,359} = 33.4$ | **< 0.001** |

Notes: Linear mixed models including the identity of the mating group as a random effect were run separately for each sex. Response variables ($I_T$, $I_N$, $I_P$) were log-transformed to meet model assumptions, and predictors (Day, Day$^2$) were centred to facilitate interpretation of fixed effects coefficients. Statistically significant results ($p < 0.05$) are in bold. F-statistics, *p*-values and degrees of freedom were calculated using the Satterthwaite approximation.

variance-based metrics due to insufficiently long sampling of mating and parentage data. Our approach suggests that sampling durations substantially shorter than the length of a selective window (e.g. mating season or breeding event) will result in overestimations. Practically, prolonged sampling for entire reproductive periods, particularly in species without clearly demarcated reproductive events (e.g. defined breeding seasons), will be challenging. In such cases appropriate sampling durations could be determined empirically by estimating the length of time required for opportunity estimates to reach an asymptote, or statistically controlling for the duration of sampling. Our results also suggest future cross-study comparisons of measures of the opportunity for sexual selection should seek to control for variation in the sampling duration and frequency of sampling.

Despite the limitations associated with measuring opportunity instantaneously, we identify a potential utility in combining cumulative and shorter-term instantaneous approaches to gather insights over how changes in mating behaviour may contribute to dynamic changes in the opportunity for sexual selection. While cumulative opportunity estimates provide the most accurate representation of the maximum potential strength of sexual selection over a breeding event, daily instantaneous estimates (or otherwise highly temporally restricted instantaneous estimates) may help identify the periods or conditions under which mating patterns change. For example, while intrasexual competition and non-random mating may slow the erosion of $I_M$ over

time, instantaneous measures over shorter time-windows may allow researchers to identify the conditions or seasonal periods that correlate with increases in non-random trends in cumulative opportunity measures. For example, shorter-term instantaneous measures may indicate an increase in variance in reproductive success over time, suggesting sustained decreases in the cumulative opportunity for selection may be accelerated by shifts in patterns of competition over time (e.g. via alternative tactics[9,70], environmental conditions[18,71], and mating preferences[72,73]) versus simply due to the accumulation of repeated iterations of broadly similar mating patterns. When accompanied with appropriate null models of random mating, such instantaneous and cumulative comparisons may give insights over how changes in mating behaviour may contribute to dynamic changes in opportunities for sexual selection.

In summary, our results reveal that estimates of opportunity for pre- and postcopulatory sexual selection can drastically decrease over short time scales and that this decrease can be at least partially explained by cumulative patterns of random mating alone, but also indicate that intrasexual competition contributes to slow the decrease over time. We show that mating data collected over short snapshots of time will overestimate the opportunity for sexual selection, particularly if sampling is conducted towards the end of a reproductive period. Collecting longitudinal data should mitigate this issue provided the sampling period is sufficiently long to detect the cumulative effect of intrasexual competition. Finally, our work highlights the pitfalls of using variance-based metrics as surrogate measures of sexual selection, particularly over highly restricted time periods, and indicate that simulations are required to disentangle temporal variation in the opportunity for sexual selection driven by stochastic and biological processes.

## Methods
All analyses were conducted using R v.3.6.2[74]. Linear models were run using base R, and linear mixed models (LMMs) were run using the package 'lme4' v1.1-30[75]. F-statistics, p-values and degrees of freedom for LMMs were calculated using the Satterthwaite approximation implemented in the package 'lmerTest' v3.1-3[76].

### Calculating opportunities for sexual selection
We collected data from the literature encompassing seven studies totalling 60 mating datasets across seven species in both natural and artificial settings. Datasets included water striders[8] (*Aquarius remegis*), howler monkeys[77] (*Alouatta caraya*), squirrel monkeys[66] (*Saimiri oerstedi*), strawberry poison-dart frogs[78] (*Dendrobates pumilio*), Hawaiian swordtail crickets[79] (*Laupala cerasina*), jackdaws[80] (*Corvus monedula*) and red junglefowl, (*Gallus gallus*)[56,67,68,81]. We used datasets from a set of studies that had been compiled by a previously published review, which conducted a literature search to examine the mating patterns of multiple species[82]. Briefly, this previous review conducted multiple searches on Web of Science aimed at locating published behavioural data of mating groups that provided information on which males copulated with which females, required to calculate temporal patterns in opportunity for precopulatory selection. A subset of studies was not included because they only provided temporal mating data for a minority of the observed males and females. We further supplemented and updated this initial set of studies, by conducting updated versions of the searches on Web of Science spanning from the time of the original search (14th February 2017) until the 19th March 2021 (i.e. searches were limited to 2017–2021). Search terms were tailored to locate studies with behavioural mating data or that reported sexual selection metrics that required the appropriate raw mating data.

The first search used the TOPIC field and contained the following search terms ("Bateman* gradient*") OR ("Bateman* slope*") OR ("Bateman* principle*") OR ("opportunity* for selection") OR ("opportunity* for sexual selection") AND ("Sexual selection").

This resulted in 35 records. The second search included the TOPIC terms (sexual network* OR social network*) AND (sexual selection OR mating system). This search returned 234 records. The third search contained the TOPIC terms (mating* or copulat*) AND (behavio*) AND (observ*), and was restricted to the journals Animal Behaviour, Behavioral Ecology, and Behavioral Ecology and Sociobiology. This returned 100 records. Searches spanned the following Web of Science indexes: SCI-EXPANDED, SSCI, A&HCI, CPCI-S, CPCI-SSH, BKCI-S, BKCI-SSH, ESCI, CCR-EXPANDED, IC. Studies were selected on the basis that they provided clear information on which males copulated with which females over successive days, which is required to calculate temporal patterns in opportunity for precopulatory selection. Studies with group sizes of only two males or females, or where individuals were not allowed to freely interact and mate with each other (e.g. repeated experimental pairs) were not included. Similarly, studies relying solely on molecular parentage to infer mating data with no behavioural mating observations were not included (for full details of each study, see Supplementary Table 1). Time spans over which data was collected within the studies varied between 6 and 150 days, where the first day of the available dataset was treated as day 1. For each dataset mating was inferred either behaviourally – or in the case of red junglefowl – through a combination of behaviour and genetic parentage analysis where pairs that were not observed mating, but produced offspring together, were assumed to have copulated. Individuals that did not copulate were allocated an M of zero[52–54].

For each dataset we calculated the opportunity for precopulatory selection ($I_M$) over successive days for males and females across all species. $I_M$ is the standardised variance in mating success ($I_M = \sigma_M^2/\bar{M}^2$, where M = the number of mating partners) - a widely used index estimating the maximum potential strength of precopulatory sexual selection[3,25,47,83]. For the red junglefowl dataset we utilised parentage data containing the order, lay day and parentage of each fertilised egg, to assess temporal patterns in additional opportunity estimates namely: the standardised variance in average partners' fecundity ($I_N$), the opportunity for postcopulatory sexual selection ($I_P$), and the opportunity for total sexual selection ($I_T$). Opportunity estimates are calculated as $I_x = \sigma_x^2/\bar{x}^2$ where x is the given measure of reproductive success or its components i.e. number of offspring (T), the mean fecundity of male mating partners (N) and male paternity share (P).

We calculated opportunities over successive days in two ways, (1) cumulatively, where estimates considered reproductive patterns from all preceding days; and (2) instantaneously, where estimates were considered independently on each day. In all species cumulative M was calculated as the number of unique mating partners up to and including that day, and therefore cumulative $I_M$ represents the maximum possible strength of selection over a sampling period up to and including a given day. Daily (i.e. instantaneous) calculations of $I_M$ on the other hand, only included unique mating partners (M) on each separate day. Therefore, consistent changes in daily instantaneous $I_M$ would suggest that patterns of mating on individual days themselves shift over-time (e.g. towards more or less egalitarian share of mating success) rather than changes in $I_M$ resulting from the daily accumulation of consistent mating patterns.

In red junglefowl cumulative reproductive success (T) was calculated as the total number of embryos assigned to males and females up to a given day, while daily T was calculated as the number of embryos assigned to an individual on each individual day. Cumulative male paternity share (P) was calculated as the percentage of embryos sired by a male across all the eggs laid by females he successfully mated with on the previous days, while daily P was calculated as the percentage of embryos sired by a male on a single day. Cumulative N was calculated as the average number of embryos produced by all the females a male successfully mated with during previous days[49,84], while daily N refers to the average number of embryos generated by these females on a given day. Given that in red junglefowl the sperm from an insemination is stored within female sperm storage tubules for up to 14 days, copulations on the first day of a trial can be assumed to compete for fertilisation of ova across all subsequent days[68,85]. Males that failed to copulate successfully were excluded from calculations of P, whereas males that successfully copulated but sired no offspring received zero P. Additionally, if a male mated successfully with a female for the first time on the day she laid an egg, this egg did not enter the calculation of his P or N since his sperm were unlikely to fertilise that egg[85].

## Comparing temporal patterns in cumulative and instantaneous opportunities for sexual selection

We evaluated temporal trends in $I_M$ and the impact of cumulative versus daily instantaneous approaches for each species and sex separately. We used linear regression analyses for studies without replicate groups, and linear mixed-effects models (LMMs) including a random effect of group identity for studies with more than one replicate group. We included $I_M$ as the response variable with day of the mating trial (continuous), the sampling method (2-level factor, cumulative vs. daily), and their interaction as predictors. For the red junglefowl dataset we ran additional LMMs including the different opportunity indices (i.e. $I_T$, $I_N$, and $I_P$) as response variables, and mating groups ($n = 20$) as random effects. When models violated the assumption of homoscedasticity, responses (i.e. $I_M$, $I_N$, $I_P$, $I_T$) were log-transformed. Moreover, in many models relationships were curvilineal despite transformations, so for these models we included a quadratic effect of day order (Day$^2$) and its interaction with the sampling method (Day$^2 \times$ Method). Predictors were centred around their means to facilitate interpretation of model coefficients[86].

Because in experimental studies day 1 corresponds to when sexually isolated individuals are first introduced to the opposite sex, several sexual behaviours may change drastically over the first days as initial sexual novelty and mating propensity are replaced by familiarity and resistance/choosiness[87,88]. To investigate this potentially confounding effect and assess the robustness of temporal patterns in cumulative $I_M$, we ran 1000 simulations swapping the day sequence arbitrarily (code available in figshare[89]). If mating behaviour on day 1 is the main factor driving temporal patterns in $I_M$, we expect results across simulations to vary drastically in slope.

## Testing for deviations from null expectations based on random matings and fertilisations

We assessed whether observed temporal trends in the opportunity for precopulatory sexual selection ($I_M$) across males and females deviated from trends calculated from simulations assuming random mating for all species. For red junglefowl we additionally compared temporal trends in male cumulative opportunity for selection on partner fecundity ($I_N$) and paternity share ($I_P$), and male and female total opportunity for sexual selection on reproductive success ($I_T$) against trends generated from simulations assuming random fertilisations.

For each species we ran 10,000 random mating simulations using custom scripts (code available in figshare[89]) with the following assumptions: (i) individuals mate randomly over a reproductive period, with the sex ratio and group size being equal to those reported by a study or extrapolated from its dataset, (ii) the total number of mating events on each day and for each individual is an integer value[51] and equal to the number of mating events in the dataset, and (iii) all individuals are assumed to be present – and available to mate – on each day of the trial. For the red junglefowl datasets we additionally simulated random fertilisations which included two extra assumptions: (iv) the probability each female will lay an egg is equal to the average laying probability across females in the empirical data, which was calculated independently for each day of the trial and only included fertilised eggs, and (v) all males that copulated with a female at least one day before her laying an egg had an equal probability of fertilising that egg. For simplicity, the number of copulations of a male with the same

female did not affect his probability of fertilising an egg. Calculations of simulated cumulative opportunity indices (i.e. $I_M$, $I_N$, $I_P$, $I_T$) were performed as described above. We compared predicted means and 95% confidence intervals of observed $I_T$, $I_M$ $I_N$, and $I_P$ with the 95% range of the simulated values. Days in which mean observed values and confidence intervals fell outside the 95% range of simulated values were considered to be significantly different from randomised values[90,91].

Comparing observed data against null expectations is particularly important because opportunity indices have been criticised for being unable to distinguish variance arising from competitive and stochastic processes[42,43,50]. Therefore, simulations seek to identify possible temporal trends that are caused by factors other than sexual selection. For example, $I_M$ should decrease over time as individuals tend to progressively mate with a larger proportion of the available members of the opposite sex, thus achieving a similarly high M (i.e. given enough time, all individuals of a population can mate with all available partners exhausting intrasexual variation in M). Similarly, $I_P$ may also decrease over time due to variation in P decreasing as a function of sample size (i.e. given enough eggs, all males mating with a female ought to sire some embryos). Therefore, assessing whether observed cumulative opportunity indices exceed null expectations elucidates whether variance in some components of reproductive success is higher than expected by chance, consistent with the signature of sexually selected traits or strategies.

### Reporting summary

Further information on research design is available in the Nature Portfolio Reporting Summary linked to this article.

## Data availability

The data generated in this study have been deposited in the figshare database under accession code https://doi.org/10.6084/m9.figshare.21902133.v1[89]. Raw mating data used to generate the opportunity estimates can be found at: Hawaiian swordtail crickets[79] (*Laupala cerasina*): https://doi.org/10.5061/dryad.9jd86, jackdaws[80] (*Corvus monedula*): https://doi.org/10.5061/dryad.j0zpc868z, red junglefowl[56,66,67,81] (*Gallus gallus*): https://doi.org/10.6084/m9.figshare.21902133.v1, water striders[8] (*Aquarius remegis*): https://doi.org/10.5061/dryad.rq56t, squirrel monkeys[66] (*Saimiri oerstedi*): Table 2 in the original paper, howler monkeys[77] (*Alouatta caraya*): Table 2 in the original paper, strawberry poison-dart frogs[78] (*Dendrobates pumilio*): Appendix 1 in the original paper. Source data are provided with this paper.

## Code availability

Custom R scripts for random mating simulations and shuffling of the day order have been deposited in the figshare database under accession code https://doi.org/10.6084/m9.figshare.21902133.v1[89].

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

## Acknowledgements

R.C. was supported by a DPhil scholarship from the Brazilian National Council for Scientific and Technological Development (CNPq) (grant no. 234988/2014-2), G.C.M. was supported by a Ph.D. CASE scholarship from the Biotechnology and Biological Sciences Research Council and Aviagen Ltd, an industrial LINK award from the Biotechnology and Biological Sciences Research Council and Aviagen Ltd (grant no. BB/L009587/1) to T.P., and by the National Research, Development and Innovation Office, Hungary (grant no. FK 134741). D.S.R. was supported by a research grant from the Natural Environment Research Council (grant no. NE/H006818/1). T.P. was supported by a research grant from the Natural Environment Research Council (grant no. NE/H008047/1), an industrial LINK award from the Biotechnology and Biological Sciences Research Council and Aviagen Ltd (grant no. BB/ L009587/1), and a research grant from the Biotechnology and Biological Sciences Research Council (grant no. BB/V001256/1).

## Author contributions
R.C., G.C.M., and T.P. conceived the study. G.C.M. conducted the fieldwork and led the literature search. R.C. analysed the data, developed the simulations, and produced the figures. D.S.R. performed the molecular work and analyses for parentage assignment. R.C., T.P., and G.C.M. wrote the manuscript, with input from D.S.R. All authors gave final approval for publication.

## Funding

## Competing interests
The authors declare no competing interests.
