## [Peer Review File · Nature Communications]

Disentangling the causes of temporal variation in the opportunity for sexual selectionREVIEWER COMMENTS

Reviewer #1 (Remarks to the Author):

In this study, the authors explore how the potential for sexual selection varies temporally across multiple species. They address this question by quantifying temporal changes in the opportunity for pre-copulatory sexual selection over cumulative days in males and females for several species, and they additionally explore the extent to which any changes in the opportunity for sexual selection are related to stochastic versus deterministic processes using null models. The use of multiple species and null models is a particularly powerful approach.

The results of this study suggest that the opportunity for sexual selection decreases across days with respect to pre-copulatory sexual selection in six of the seven species considered for both sexes. In general, the opportunity for sexual selection deviated from what we would expect based on random mating in some, but certainly not all, cases. In the red junglefowl, the cumulative opportunity for total selection decreased over time in both males and females, and daily instantaneous measures led to a significant overestimation of most opportunity estimates in both sexes. Collectively, these results highlight how variation in mating success and the potential for sexual can be sensitive to both random and deterministic processes, time, sampling approach, and species-specific biology.

This is a fascinating, data-rich study on an important topic: temporal variability in the variance in mating success that is expected to relate to and influence sexual selection. The finding that the opportunity for sexual selection estimates are overestimated if calculated over small periods of time is particularly important and is likely to influence the interpretation of data in studies that quantify the opportunity for sexual selection. Likewise, the finding that observed temporal decreases in the opportunity for sexual selection are in many cases similar to what is expected under random mating is also really fascinating.

In general, this is an excellent, well-written manuscript on an important topic. I have very few comments, and I think that this work will make a substantial contribution to the field of sexual selection and evolutionary ecology in general.

Comments:

The authors note that mating matrices did not fully saturate in any of the species in the present study (L 184). However, the opportunity for sexual selection will co-vary with sex ratio biases even when deterministic factors occur because individuals can't have a fraction of a mate (i.e., some stochasticity is always associated with variation in mating success since mate number must be an integer; ref. 24 in the current manuscript). Is this accounted for in the null models? If so, this is likely worth noting explicitly (and briefly) somewhere in the text.

L 277-278: Some details of why these studies were chosen should be included in the main body of the manuscript.

L147, 'in females observed $I(\text{sub}T)$ did deviate from null expectations at any point in the trial (Fig. 2a)': I'm confused by this statement. Based on Figure 2A, it looks like the confidence intervals overlap with the expectations based on the null model for some, if not all, time points. Is this correct or am I misinterpreting something?

L234-259: This is a really thoughtful and well-written section. As someone who has been critical of using variance-based measures of mating success as proxies for sexual selection, I think that we really need more studies like the current one. Comparing the opportunity for sexual selection to what's expected under null/stochastic versus deterministic scenarios is a powerful and interesting approach.

Reviewer #2 (Remarks to the Author):

Review of Carleial et al.

This is a study of fluctuations in a widely used but flawed estimator of sexual selection, I_s : the opportunity for selection. To quote Klug et al. (2010) who conducted a thorough test of indirect sexual selection estimators: " I_s reflects the maximum, but not necessarily the realized, strength of sexual selection", " I_s only accurately predict[s] sexual selection under a limited set of circumstances" and " I_s is expected to produce spurious results that lead to the false conclusion that sexual selection is strong when it is actually weak". The authors of the current study show that I_s (the opportunity for selection) fluctuates tremendously over sampling periods and that this measure at times (for some taxa) cannot be distinguished from random mating. I think the paper is another nail in the coffin of I_s .

The authors do not frame the work in this manner however, but instead talk more about fluctuations in selection over short temporal time scales which they purport to show, but just to remind them again, I_s is only a measure of the opportunity for selection, so they are not actually showing that selection is varying at all – if you want to measure selection, then measure it (fitness plotted against phenotype). The fact that selection has not been measured means the authors cannot meet the stated aim of the paper which is to understand temporal fluctuations in sexual selection.

However, I think the authors do a great job of showing why I_s should not be used as a selection surrogate and as such the work is great additional foil to the Klug et al. (2010) paper, and for this reason I think it is important. So based on my prejudices, I think the paper should be reworked in this light. At present all it says is that an indirect and inaccurate (except under restricted circumstances) measure of sexual selection that we should not really widely use fluctuates a lot. However, as noted above this does not get to the nub of the question with respect to actual fluctuations in selection, but it does point out to additional perils of using I_s – the metric is highly variable over time. I hope that is more or less clear.

My suggestion would be a rewrite and re-frame in light of Klug et al. (2010). This would be of much great benefit to the field and would probably generate more impact too. As an aside, it was pretty remarkable that the Klug paper was cited so late in the manuscript when it has such a fundamental impact on everything associated with your work.

Minor comments:

Line 13: not clear why this statement is true? Won't mating patterns determine the strength of sexual selection rather than vice versa?

Line 45: aren't other scales "biologically relevant" or are you implying that shorter time periods are also relevant?

Lines 67-69: to understand temporal variation in sexual selection you actually need to measure sexual selection over time – fitness/phenotype Lande-Arnold-Wade measures. Understanding how I_s varies will not tell us if selection itself is varying because I_s does not measure selection.

Line 176-178: no this is not true Klug et al (2010) show why/how. To cite a much older and well and truly superseded paper sets up a strawman here.

Line 184, 203, 223 and 236-238: all provide statements showing how poorly I_s behaves and while noting that you finally state this (lines 236-8), the preceding statements should also be framed in the same way as the latter.

Table 1. The legend needs to more thoroughly explain what the table actually shows.

I_s is the right hand column instantaneous and for example, what do the interaction terms denote. Put it there otherwise the reader has to jump back and forth from the main text to the table to try and understand what is going on.

There are some formatting errors in the references.

I hope this review was useful.

Klug H, Heuschele J, Jennions MD & Kokko H. 2010. The mismeasurement of sexual selection. *J Evol Biol* 23:447-462.

RESPONSE TO REVIEWERS' COMMENTS

We appreciate the positive and constructive comments from both reviewers. In the point-by-point response below, we explain in detail how we have addressed each comment in turn (our responses in bold). Novel additions and modifications to the previous version of the manuscript's text are highlighted in yellow in the revised manuscript. Finally, we have modified Figures 1 and 2 as follows:

-Figure 1 now shows confidence intervals rather than standard errors. The Result section has been amended to reflect this correction i.e. the statement about differences in patterns between male and female howler monkey was removed (previously on lines 123-126).

-For consistency and to conform with Nature comm. artwork requirements, Figure 2 colour codes now match those of Figure 1 (i.e. green and purple).

Reviewer #1 (Remarks to the Author):

In this study, the authors explore how the potential for sexual selection varies temporally across multiple species. They address this question by quantifying temporal changes in the opportunity for pre-copulatory sexual selection over cumulative days in males and females for several species, and they additionally explore the extent to which any changes in the opportunity for sexual selection are related to stochastic versus deterministic processes using null models. The use of multiple species and null models is a particularly powerful approach.

The results of this study suggest that the opportunity for sexual selection decreases across days with respect to pre-copulatory sexual selection in six of the seven species considered for both sexes. In general, the opportunity for sexual selection deviated from what we would expect based on random mating in some, but certainly not all, cases. In the red junglefowl, the cumulative opportunity for total selection decreased over time in both males and females, and daily instantaneous measures led to a significant overestimation of most opportunity estimates in both sexes. Collectively, these results highlight how variation in mating success and the potential for sexual selection can be sensitive to both random and deterministic processes, time, sampling approach, and species-specific biology.

This is a fascinating, data-rich study on an important topic: temporal variability in the variance in mating success that is expected to relate to and influence sexual selection. The finding that the opportunity for sexual selection estimates are overestimated if calculated over small periods of time is particularly important and is likely to influence the interpretation of data in studies that quantify the opportunity for sexual selection. Likewise, the finding that observed temporal decreases in the opportunity for sexual selection are in many cases similar to what is expected under random mating is also really fascinating.

In general, this is an excellent, well-written manuscript on an important topic. I have very few comments, and I think that this work will make a substantial contribution to the field of sexual selection and evolutionary ecology in general.

R: We are pleased that this reviewer enjoyed the manuscript, and we are grateful for their positive comments.

Comments:

The authors note that mating matrices did not fully saturate in any of the species in the present study (L 184). However, the opportunity for sexual selection will co-vary with sex ratio biases even when deterministic factors occur because individuals can't have a fraction of a mate (i.e., some stochasticity is always associated with variation in mating success since mate number must be an integer; ref. 24 in the current manuscript). Is this accounted for in the null models? If so, this is likely worth noting explicitly (and briefly) somewhere in the text.

R: This is a very good point. Sex ratios in the simulations were identical to the sex ratios in the original studies, as were the number of matings on each day. For example, if 20 matings occurred on the first day of a study, males and females were randomly paired 20 times in the simulations. Therefore, the number of matings was always an integer and accounts for the stochasticity suggested by the reviewer. We have added a relevant reference which contemplates the reviewer's point (ref 51, Klug and Stone 2021), and noted this explicitly (and briefly) as suggested in the Methods section as below:

Lines 374-376: "(i) individuals mate randomly over a reproductive period, with the sex ratio and group size being equal to those reported by a study or extrapolated from its dataset, (ii) the total number of mating events on each day and for each individual is an integer value⁵¹ and equal to the number of mating events in the dataset."

L 277-278: Some details of why these studies were chosen should be included in the main body of the manuscript.

R: We agree. The following statements have been added to the main body:

Lines 293-302: "We used datasets from a set of studies that had been compiled by a previously published review, which conducted a literature search to examine the mating patterns of multiple species⁸². We further supplemented and updated this initial set of studies, by conducting new systematic searches on the Web of Science (see Supplementary Material for full details). Studies were selected on the basis that they provided clear information on which males copulated with which females over successive days, which is required to calculate temporal patterns in opportunity for precopulatory selection. Studies with group sizes of only two males or females, or where individuals were not allowed to freely interact and mate with each other (e.g., repeated experimental pairs) were not included. Similarly, studies relying solely on molecular parentage to infer mating data with no behavioural mating observations were not included (for full details, including of each study, see Supplementary Material and Supplementary Table 1)."

L147, 'in females observed $I(\text{sub}T)$ did deviate from null expectations at any point in the trial (Fig. 2a)': I'm confused by this statement. Based on Figure 2A, it looks like the confidence intervals overlap with the expectations based on the null model for some, if not all, time points. Is this correct or am I misinterpreting something?

R: Thanks for pointing this out. This was no misinterpretation on the side of the reviewer, but a typo on our side. The statement should have been negative: "in females

observed I_T did not deviate from null expectations...". This has been corrected. Line 147

L234-259: This is a really thoughtful and well-written section. As someone who has been critical of using variance-based measures of mating success as proxies for sexual selection, I think that we really need more studies like the current one. Comparing the opportunity for sexual selection to what's expected under null/stochastic versus deterministic scenarios is a powerful and interesting approach.

R: We are delighted that the reviewer enjoyed the manuscript and found our approach useful.

Reviewer #2 (Remarks to the Author):

This is a study of fluctuations in a widely used but flawed estimator of sexual selection, I_s : the opportunity for selection. To quote Klug et al. (2010) who conducted a thorough test of indirect sexual selection estimators: " I_s reflects the maximum, but not necessarily the realized, strength of sexual selection", " I_s only accurately predict[s] sexual selection under a limited set of circumstances" and " I_s ... is expected to produce spurious results that lead to the false conclusion that sexual selection is strong when it is actually weak". The authors of the current study show that I_s (the opportunity for selection) fluctuates tremendously over sampling periods and that this measure at times (for some taxa) cannot be distinguished from random mating. I think the paper is another nail in the coffin of I_s . The authors do not frame the work in this manner however, but instead talk more about fluctuations in selection over short temporal time scales which they purport to show, but just to remind them again, I_s is only a measure of the opportunity for selection, so they are not actually showing that selection is varying at all – if you want to measure selection, then measure it (fitness plotted against phenotype). The fact that selection has not been measured means the authors cannot meet the stated aim of the paper which is to understand temporal fluctuations in sexual selection.

R: We agree with the reviewer that I_s (denoted I_M in the manuscript) does not represent actual sexual selection, which should be measured as selection gradients. Throughout the previous version of the manuscript, we had strived to emphasize that I_M captures the potential for, not the actual strength of sexual selection. However, we agree with the reviewer that the limitations of opportunity estimates are of paramount importance and should be stated more clearly. Moreover, we agree that a key result emerging from our manuscript is to show that it is misleading to interpret temporal changes in I_M as temporal changes in sexual selection. Therefore, we have changed the title of our manuscript to better reflect our results, and have more clearly emphasised that our goal was to assess the opportunity for sexual selection, not actual sexual selection. We have further emphasised that I_M does not represent actual sexual selection throughout other sections of the manuscript as follows:

Title: "Temporal variation in the opportunity for sexual selection: disentangling methodological artifacts from biological causes"

In the Abstract:

Lines 24-25: “Collectively, we show that variance-based metrics of selection change rapidly, are highly sensitive to sampling durations, and likely lead to substantial misinterpretation if used as indicators of sexual selection.”

In the Introduction:

Lines 47-49: “Thus, a widely used alternative approach estimates the potential for – rather than the strength of – sexual selection in a population as standardised intrasexual variance in reproductive success.”

Lines 56-59: “The value of these variance-based metrics however, remains debated, largely due to their inability to distinguish between variation that arises due to intrasexual competition and variation in reproductive success due to stochastic processes^{42,43,50,51}. For example, Klug et al.⁴² showed that the number of unmated individuals tends to increase as the operational sex ratio becomes more biased, so the opportunity for sexual selection will also increase even if individuals mate at random.”

Lines 67-70: “However, because the opportunity for sexual selection only measures the maximum potential for sexual selection and does not discriminate between competitive and stochastic processes^{42,43,51}, systematic changes of this metric over time may be mistakenly interpreted as changes in actual sexual selection.”

In the Discussion:

Lines 174-176: “These results reinforce previous suggestions that the opportunity for sexual selection should not be used as a proxy of actual sexual selection, but that the use of null models assuming random mating may help mitigate potential issues.”

Lines 202-205: “More importantly, such comparisons across different mating systems should include longitudinal measures of actual selection gradients⁴² to explore how net patterns of phenotypic sexual selection are shaped by cumulative patterns of mating over time, such as via changes in female selectivity or alternative mating tactics.”

Lines 235-236: “We also identified other concerns associated with using I_M as a surrogate for the strength of sexual selection.”

Line 253: “Despite the limitations associated with measuring opportunity instantaneously, [...]”

Lines 267-269: “When accompanied with appropriate null models of random mating, such instantaneous and cumulative comparisons may help to gather insights over how changes in mating behaviour may contribute to dynamic changes in opportunities for sexual selection.”

Lines 277-280: “Finally, our work highlights the pitfalls of using variance-based metrics as surrogate measures of sexual selection, particularly over highly restricted time

periods, and indicate that simulations are required to disentangle temporal variation in the opportunity for sexual selection driven by stochastic and biological processes.”

However, I think the authors do a great job of showing why I_S should not be used as a selection surrogate and as such the work is great additional foil to the Klug et al. (2010) paper, and for this reason I think it is important. So based on my prejudices, I think the paper should be reworked in this light. At present all it says is that an indirect and inaccurate (except under restricted circumstances) measure of sexual selection that we should not really widely use fluctuates a lot. However, as noted above this does not get to the nub of the question with respect to actual fluctuations in selection, but it does point out to addition perils of using I_S – the metric is highly variable over time. I hope that is more or less clear. My suggestion would be a rewrite and re-frame in light of Klug et al. (2010) (reference 42 in the text). This would be of much great benefit to the field and would probably generate more impact too.

R: We thank the reviewer for pointing out the importance of our manuscript. While we believe opportunities can still be useful if paired with simulations and interpreted with its limitations in mind, we agree with the reviewer that I_M (I_S) is widely – and incorrectly interpreted – as an estimate of actual sexual selection. We have therefore more strongly emphasized the issues related to I_M – and how our results relate to and demonstrate these issues – in several places in the manuscript (see our response to the point above). We also included statements framed with respect to the Klug et al. (2010) paper as suggested by the reviewer:

Lines 57-59: “For example, Klug et al.⁴² showed that the number of unmated individuals tends to increase as the operational sex ratio becomes more biased, so the opportunity for sexual selection will also increase even if individuals mate at random.”

Lines 202-205: “More importantly, such comparisons across different mating systems should include longitudinal measures of actual selection gradients⁴² to explore how net patterns of phenotypic sexual selection are shaped by cumulative patterns of mating over time, such as via changes in female selectivity or alternative mating tactics.”

Lines 220-222: “This scenario, in which mate monopolisation is substantial, has been previously identified as one of the few cases where opportunity estimates may correlate with actual sexual selection⁴².”

Lines 277-280: “Finally, our work highlights the pitfalls of using variance-based metrics as surrogate measures of sexual selection, particularly over highly restricted time periods, and indicate that simulations are required to disentangle temporal variation in the opportunity for sexual selection driven by stochastic and biological processes.”

Moreover, the paragraph discussing the limitations of I_M , as well as the importance of null models (both of which were flagged by Klug et al. 2010 and Jennions et al. 2012), was moved to earlier in the discussion to give more prominence to this important issue (lines 206-234, this paragraph used to be in lines 234-259).

As an aside, it was pretty remarkable that the Klug paper was cited so late in the manuscript when it has such a fundamental impact on everything associated with your work.

R: We whole-heartedly agree. Upon correcting a few reference formatting errors (see comment below) we realised an earlier reference to Klug et al. 2010 was absent. We apologise and are very thankful to the reviewer for pointing out the issue. All references are now fixed, and the Klug et al. 2010 paper (ref 42) is now cited in the second paragraph of the introduction and in all other relevant sections. In addition, we have also included a more recent reference from the same research group exploring biases in opportunities (Klug and Stone 2022).

Minor comments:

Line 13: not clear why this statement is true? Won't mating patterns determine the strength of sexual selection rather than vice versa?

R: We agree the causality here could be reversed. We have removed this statement as part of editing the abstract.

Line 45: aren't other scales "biologically relevant" or are you implying that shorter time periods are also relevant?

R: We meant that short time periods are still relevant, despite being short. We amended the sentence to make our point clearer:

Lines 41-42: “..less is known about the potential for rapid fluctuations in sexual selection over these much shorter, but still biologically relevant, scales.”

Lines 67-69: to understand temporal variation in sexual selection you actually need to measure sexual selection over time – fitness/phenotype Lande-Arnold-Wade measures. Understanding how I_s varies will not tell us if selection itself is varying because I_s does not measure selection.

R: We agree with the reviewer, but our aim was precisely to investigate whether temporal patterns in opportunities could in principle accurately reflect temporal patterns in sexual selection, and whether randomised null models as advocated by e.g. Jennions et al. (2012) can help make sense of these temporal patterns by comparing empirical estimates with estimates calculated from null expectations. This remains important given the continued use of opportunities in sexual selection studies (e.g. Janicke & Morrow 2018, Cattelan et al. 2020, Mokos et al. 2021). However, we can see how the statement highlighted by the reviewer did not properly convey our point. We have therefore added a new sentence to better reflect the reviewer's observation:

Lines 67-70: “However, because the opportunity for sexual selection only measures the maximum potential for sexual selection and does not discriminate between competitive and stochastic processes^{42,43,51}, systematic changes of this metric over time may be mistakenly interpreted as changes in actual sexual selection.”

Line 176-178: no this is not true Klug et al (2010) show why/how. To cite a much older and well and truly superseded paper sets up a strawman here.

R: Thanks for pointing this out. We agree that this statement will not be true unless a few other assumptions are met. We have removed the statement to avoid any potential misunderstanding and maintain brevity.

Line 184, 203, 223 and 236-238: all provide statements showing how poorly I_S behaves and while noting that you finally state this (lines 236-8), the preceding statements should also be framed in the same way as the latter.

R: We now clearly state that our results show how I_M (I_S) is a poor proxy for sexual selection in the first paragraph of our discussion. Moreover, as part of modifying the text throughout we have put a much greater emphasis on the issues associated with the use of I_M (see above). However, we have been careful where possible to avoid repetition and incorrectly suggesting that any temporal change in I_M is in principle incorrect. Further changes include edits to lines 203 and 223 as suggested by the reviewer:

Lines 235-238: “We also identified other concerns associated with using I_M as a surrogate for the strength of sexual selection. Our comparisons between cumulative and daily instantaneous measures of opportunity for selection indicate that even small differences in sampling periods of a few days can lead to very different conclusions regarding the potential for sexual selection.”

Lines 253-256: “Despite the limitations associated with measuring opportunity instantaneously, we identify a potential utility in combining cumulative and shorter-term instantaneous approaches to gather insights over how changes in mating behaviour may contribute to dynamic changes in the opportunity for sexual selection.”

Table 1. The legend needs to more thoroughly explain what the table actually shows. Is the right hand column instantaneous and for example, what do the interaction terms denote. Put it there otherwise the reader has to jump back and forth from the main text to the table to try and understand what is going on.

R: We thank the reviewer for pointing out a lack of clarity. We have now amended the table legend and predictor names to make it clear what the factor levels are and what the interactions denote.

There are some formatting errors in the references.

R: Corrected. Thank you.

I hope this review was useful.

R: We believe the changes suggested by the reviewer have considerably strengthened the manuscript, and also allowed us to better underline the key strengths of our approach.

Literature cited:

Cattelan, S., Evans, J.P., Garcia-Gonzalez, F., Morbiato, E. and Pilastro, A. Dietary stress increases the total opportunity for sexual selection and modifies selection on condition-dependent traits. *Ecol. Lett.* 23, 447-456 (2020).

Janicke, T. and Morrow, E.H. Operational sex ratio predicts the opportunity and direction of sexual selection across animals. *Ecol. Lett.* 21, 384-391 (2018).

Jennions, M. D., Kokko, H. and Klug, H. The opportunity to be misled in studies of sexual selection. *J. Evol. Biol.* 25, 591–598 (2012).

Klug H., Heuschele J., Jennions M.D. and Kokko, H. The mismeasurement of sexual selection. *J. Evol. Biol.* 23:447-462 (2010).

Klug, H. and Stone, L. More than just noise: Chance, mating success, and sexual selection. *Ecol. Evol.* 11, 6326–6340 (2021).

Mokos, J., Scheuring, I., Liker, A., Freckleton, R.P., and Székely, T. Degree of anisogamy is unrelated to the intensity of sexual selection. *Sci. Rep.* 11, 1-11 (2021).

REVIEWERS' COMMENTS

Reviewer #1 (Remarks to the Author):

I was the original referee 1 of this manuscript, and I find the revised manuscript to be excellent. The authors have now addressed my previous concerns in a detailed and thorough manner. I appreciate their attention to my earlier comments. I also reviewed the responses to comments from referee 2. Referee 2 provided a number of thoughtful and detailed comments. These comments largely centered around re-framing the manuscript to 1) better acknowledge that I_s is a measure of the potential (rather than actual strength of) sexual selection, 2) more directly linking the current work to limitations of using I_s as a proxy for sexual selection, and 3) acknowledging that I_s cannot distinguish between deterministic and stochastic processes. The revisions in response to the comments from referee 2 have led to many improvements in the revised manuscript.

I have a few remaining comments, which I detail below.

L20: Minor typo here—the period should be a comma or just deleted.

L25-26: As written, I don't agree with this statement as it perhaps suggests that simulations can fully mitigate these issues. I suggest deleting this last sentence or modifying it to something such as: 'However, simulations can begin to disentangle sampling artefacts from biological mechanisms'.

L43: I suggest deleting 'ideally'. You nicely make the point that we often have to use proxies of the strength of sexual selection below, so it seems unnecessary.

L47: Agreed.

L75, 'However, currently there is little information...': Given the revisions which better highlight the limitations of I_s , it would be worth adding a sentence here to justify the importance of continuing to understand how the opportunity for sexual selection behaves over time. For example, at the end of this paragraph, you could add something such as: 'Understanding how variance-based metrics behave through time in relation to deterministic and stochastic processes is important because variance in mating success is a prerequisite for sexual selection'. I'm not sure if this is quite the best sentence, but something that highlights the utility of considering variance-based metrics would improve this paragraph. It's clear from earlier work, the comments of referee 2, my own work (e.g. Klug et al. 2010, Klug and Stone 2021), etc. that there's a lot of criticism surrounding the opportunity for sexual selection. Given this, I think that we need some mention here of why it's valuable to continue focusing on variance in mating success in any context.

L160: I suggest replacing 'in sexual selection, and its quantitative measures,' with 'mating success'.

L176: I suggest replacing 'assuming random mating may help mitigate potential issues' with 'can provide insight into how random versus deterministic processes are expected to influence mating dynamics' (or something similar).

Reviewer #2 (Remarks to the Author):

I have nothing further to add and thank the authors for a thorough revision. It is a very nice bit of work and the additional clarity on problems with the use of variance metrics is really helpful. If it were me, I'd be even tougher in the stance, but that is my personal prejudice and does not detract at all from the quality and substance of this paper.

Nice work.

We thank both reviewers for their very positive evaluation of our revised manuscript. In the point-by-point below we detail how we addressed the remaining minor comments of Reviewer 1. Our responses are in bold and reviewer comments are in normal text.

In addition we have made further minor edits in line with those requested by the Author Checklist including (i) an edited title without punctuation, (ii) a brief description of the study's main findings at the end of the introduction and (iii) removal of subheadings from the results section.

RESPONSE TO REVIEWERS' COMMENTS

Reviewer #1 (Remarks to the Author):

I was the original referee 1 of this manuscript, and I find the revised manuscript to be excellent. The authors have now addressed my previous concerns in a detailed and thorough manner. I appreciate their attention to my earlier comments. I also reviewed the responses to comments from referee 2. Referee 2 provided a number of thoughtful and detailed comments. These comments largely centered around re-framing the manuscript to 1) better acknowledge that I_s is a measure of the potential (rather than actual strength of) sexual selection, 2) more directly linking the current work to limitations of using I_s as a proxy for sexual selection, and 3) acknowledging that I_s cannot distinguish between deterministic and stochastic processes. The revisions in response to the comments from referee 2 have led to many improvements in the revised manuscript.

R: We are delighted that the reviewer agrees with the changes made to the manuscript.

I have a few remaining comments, which I detail below.

L20: Minor typo here—the period should be a comma or just deleted.

R: Corrected. Thank you.

L25-26: As written, I don't agree with this statement as it perhaps suggests that simulations can fully mitigate these issues. I suggest deleting this last sentence or modifying it to something such as: 'However, simulations can begin to disentangle sampling artefacts from biological mechanisms'.

R: Modified as suggested. Sentence now reads "However, simulations can begin to disentangle stochastic variation from biological mechanisms." Lines 25-25.

L43: I suggest deleting 'ideally'. You nicely make the point that we often have to use proxies of the strength of sexual selection below, so it seems unnecessary.

R: Agreed. Deleted as suggested.

L47: Agreed.

R: Great.

L75, 'However, currently there is little information...': Given the revisions which better highlight the limitations of I_s , it would be worth adding a sentence here to justify the importance of continuing to understand how the opportunity for sexual selection behaves over time. For example, at the end of this

paragraph, you could add something such as: 'Understanding how variance-based metrics behave through time in relation to deterministic and stochastic processes is important because variance in mating success is a prerequisite for sexual selection'. I'm not sure if this is quite the best sentence, but something that highlights the utility of considering variance-based metrics would improve this paragraph. It's clear from earlier work, the comments of referee 2, my own work (e.g. Klug et al. 2010, Klug and Stone 2021), etc. that there's a lot of criticism surrounding the opportunity for sexual selection. Given this, I think that we need some mention here of why it's valuable to continue focusing on variance in mating success in any context.

R: Excellent suggestion. We have added the sentence as suggested by the reviewer. Lines 74-76.

L160: I suggest replacing 'in sexual selection, and its quantitative measures,' with 'mating success'.

R: Modified as suggested to "mating and reproductive success". Line 161.

L176: I suggest replacing 'assuming random mating may help mitigate potential issues' with 'can provide insight into how random versus deterministic processes are expected to influence mating dynamics' (or something similar).

R: Modified as suggested. Lines 177-178.

Reviewer #2 (Remarks to the Author):

I have nothing further to add and thank the authors for a thorough revision. It is a very nice bit of work and the additional clarity on problems with the use of variance metrics is really helpful. If it were me, I'd be even tougher in the stance, but that is my personal prejudice and does not detract at all from the quality and substance of this paper.

Nice work.

R: We are delighted the reviewer appreciated our modifications to the manuscript.